# Low-Temperature Synthesis of Cu-Doped Anatase TiO_2_ Nanostructures via Liquid Phase Deposition Method for Enhanced Photocatalysis

**DOI:** 10.3390/ma16020639

**Published:** 2023-01-09

**Authors:** Mitsuhiro Honda, Tsuyoshi Ochiai, Popy Listiani, Yuma Yamaguchi, Yo Ichikawa

**Affiliations:** 1Graduate School of Engineering, Nagoya Institute of Technology, Nagoya 466-8555, Japan; 2Materials Analysis Group, Kawasaki Technical Support Department, Kanagawa Institute of Industrial Science and TEChnology (KISTEC), Kawasaki 213-0012, Japan

**Keywords:** photocatalysis, liquid phase deposition, copper doping, low-temperature synthesis

## Abstract

Titanium dioxide (TiO_2_) photocatalysis can harness the energy from sunlight, providing a solution to many green- and energy-related problems. In this study, we aimed to produce Cu doped TiO_2_ (Cu-TiO_2_) structures at a low temperature (~70 °C) under atmospheric pressure based on liquid phase deposition. The products prepared with Cu nitrate exhibited anatase-phase TiO_2_ with the presence of Cu, and the particles showed a waxberry-like structure. Changing the Cu nitrate concentration allowed control of the atomic concentration; we confirmed ~1.3 atm.% of Cu ions in the product when we applied 10 mM in the precursor solution. By doping Cu, the light absorption edge shifted to 440 nm (~2.9 eV), and we proved the photocatalytic reaction through action spectral measurement. We observed the decomposition of acetaldehyde into CO_2_ on Cu-TiO_2_ photocatalysts, which produced optimized improvements in photocatalytic activity at Cu dopant levels between 0.2 and 0.4 atm.%. This study demonstrates that the liquid phase deposition technique can be used for doping metallic ions into TiO_2_, which shows promise for preparing novel and unique nanomaterials as visible light photocatalysts.

## 1. Introduction

Photocatalysis is capable of harnessing the energy from sunlight, being a solution to many green- and energy-related problems facing the world [1,2]. Titanium dioxide (TiO_2_), in the anatase phase, is the most representative photocatalyst due to its strong oxidizing power as well as its physical and chemical stability [3,4,5,6]. The qualities of TiO_2_ have led to its diverse applications in, for example, self-cleaning surfaces, water splitting, disinfection, and air/water purification [7,8,9,10].

The trigger of photocatalysis is the absorption of photons, which are trapped by the photocatalyst to generate electron–hole pairs, subsequently inducing chemical reactions at the surface [11]. However, TiO_2_ can only respond to photons with energy larger than 3.2 eV, which is ultraviolet (UV) light, which only accounts for approximately 5% in sunlight [12]. This is considered to be one of the critical limitations for efficient photocatalysis. Thus, to date, extending the wavelength range of the photoactivation of TiO_2_ photocatalysts toward the visible-light region is a crucial task to enhance the use efficiency of solar energy.

Elemental doping is an effective strategy to increase the visible light sensitivity of TiO_2_ [13,14]. Doping can narrow the bandgap of TiO_2_ by downshifting the conduction band bottom or introducing new energy levels within the gap [15,16,17,18,19]. Such states, owing to doping ions, can be used to trap electrons or holes to separate carriers from the bands, thus allowing more carriers to successfully diffuse to the surface. Among the diverse doping elements, copper (Cu) is one of the attractive choices for visible-light harvesting and increased photocatalytic efficiency [20,21,22,23,24,25]. Although the experimental evidence for the enhancement in photocatalytic activity in TiO_2_ through copper doping is inconclusive, copper doping can modify the large bandgap of TiO_2_ to optimize its optical properties for visible-light harvesting. Mingmongkol et al. doped TiO_2_ with Cu ranging from 0.1 to 1.0 wt.%, which resulted in the bandgap decreasing from 3.20 eV for undoped TiO_2_ to 3.12 eV for 1.0 wt.% Cu-doped TiO_2_ [26]. Mathew et al. reported that the visible light absorption property of Cu-TiO_2_ increased and the band gap reduced to 2.8 eV with 0.5 mol% Cu [23]. The results of density functional theory (DFT) studies suggested that the introduction of Cu^+^ and Cu^2+^ ions creates oxygen vacancies by replacing Ti^4+^ ions in the TiO_2_ lattice. The enhancement in photocatalytic activity owing to doping Cu is driven by reduced charge carrier recombination in Cu-TiO_2_. Cu dopants (Cu^0^, Cu^+^, Cu^2+^) can create multiple bands to extend electron–hole (e^−^, h^+^) pair recombination; therefore, Cu-TiO_2_ exhibits improved electron–hole separation with photoexcitation, resulting in the increase in photocatalytic efficiency [27,28]. The purposes of doping Cu into TiO_2_ photocatalyst are: (i) modifying its large bandgap and electronic structure to optimize its optical properties for visible-light harvesting; (ii) improving each step in the charge kinetics to reduce the massive recombination of photogenerated carriers. Photocatalytic activity is optimized at Cu dopant levels between 0.1 and 2 atm.% [27,28,29,30]. Above a certain threshold, increasing the Cu dopant concentration diminishes photocatalytic activity through a combination of enhanced recombination and shading [28,30]. The diversity in the reported behavior of Cu dopants and their effects on photocatalytic activity seem to largely stem from diverse materials synthesis methods that result in Cu dopants in nonequilibrium sites as well as from the diverse approaches through which photocatalytic activity is assessed.

Cu has been added to TiO_2_ in different mole percentages for various photocatalytic applications such as hydrogen production, CO_2_ reduction, decomposition of organic contaminants, and antibacterial/bacterial killing [20,21,22,23,24,25]. Golon et al. synthesized Cu-TiO_2_ via a sol gel method; the doped sample, which was calcined at 600 °C, displayed high photocatalytic activity for phenol degradation [25]. In another study, the gel of Ti(IV) isopropoxide and Cu nitrate was nucleated for 12 h in the dark, the mass was then dried in an oven at 100 °C for 4 h, and the resulting solid mass was finally crushed and calcined at 500 °C for 3 h [31]. Among all the photocatalysts examined in the study, the sample with the lowest Cu concentration produced the most CH_4_ (1081 μL h^−1^ g^−1^) and the highest doping showed the least activity (200 μL h^−1^ g^−1^) under similar conditions. In most cases, the synthesis procedures require high-temperature calcination, so are not suitable for forming photocatalyst films on the platforms of technologically important substrates such as plastics and glasses. Another attractive approach to synthesize Cu-doped nanocrystalline TiO_2_ is the hydrothermal method; this method can be employed under self-produced pressures at low temperatures [32,33]. Typically, the precursor solution is transferred to a Teflon-lined stainless-steel autoclave and heated at ~180 °C for 12 h. The operating temperature is above the melting point of plastics, and the pressure inside a reactor is known to be in the mega pascals range; therefore, this technique is incompatible with forming a film on a flexible substrate such as plastics [34]. The precipitation method was used to synthesize Cu-doped nanocrystalline TiO_2_; this method can be performed using low-cost materials and easier manufacturing methods at industrial scales. Recently, A.M. Alotaibi et al. used aerosol-assisted chemical vapor deposition (AACVD) to deposit highly photoactive thin films of Cu-doped anatase TiO_2_ on glass substrates [22]. In their study, a solution of titanium isopropoxide, copper nitrate, and ethyl acetate was atomized using a piezoelectric device, and the generating aerosol was transferred with N_2_ flow onto the glass substrate heated at 470 °C. AACVD provides a facile route for producing highly photoactive Cu-doped TiO_2_ thin films. In such films, the interactions between substitutional and interstitial Cu in the anatase lattice can explain the extended exciton lifetimes and the enhanced photocatalytic and antibacterial ability. Thus, although many researchers have focused on the preparation of Cu-TiO_2_ using sol–gel and hydrothermal techniques, studies on synthesis or coating techniques with operation at low temperatures or under normal pressure are scarce.

In this context, in this study, we aimed at producing Cu-TiO_2_ structures at low temperature (<~80 °C). For this, we employed the liquid phase deposition (LPD) technique as a method to produce TiO_2_ nanostructures. LPD is a simple and economical approach to generate TiO_2_ with an anatase phase at room temperature, which was first reported by Deki in 1996 [35]. LPD synthesis is based on the hydrolysis of ammonium hexafluoro titanate [(NH_4_)_2_TiF_6_], with boric acid used as the fluorine ion scavenger, forming TiO_2_ via a ligand exchange reaction [35,36]. As this reaction proceeds at room temperature and atmospheric pressure, LPD is applicable for non-heat-resistant materials such as polystyrene, glass, etc. [37,38,39]. In this study, based on the LPD technique, we dissolved Cu salts into a precursor solution for Cu doping, after which we microscopically and spectroscopically characterized the products to prove the possibility of controlling the Cu-doping level.

## 2. Materials and Methods

### 2.1. Sample Preparation

Following the typical LPD procedure described in a previous study [39], we produced TiO_2_ samples through a temperature-assisted liquid phase deposition process, whose procedure is schematically illustrated in Figure 1. For the precursors, we used (NH_4_)_2_TiF_6_ (AHFT) (FUJIFILM Wako, 1st Grade) and H_3_BO_3_ (FUJIFILM Wako, 1st Grade). We separately dissolved both materials in distilled water at 0.1 M and 0.3 M for AHFT and boric acid, respectively. This ratio allowed the production of the anatase phase of TiO_2_. For doping copper ions, we then mixed the solutions with several types of Cu salts under vigorous stirring for 15 min. Here, as Cu salts, we applied Cu(NO_3_)_2_·3H_2_O (FUJIFILM Wako, Wako Special Grade), CuCl_2_ (Kojundo Chemical Laboratory Co., Ltd., Tokyo, Japan), and Cu(CH_3_COO)_2_ (FUJIFILM Wako, Wako Special Grade). Next, we fixed a glass substrate (50 × 50 mm) in a beaker so that the coating surface was kept downward to avoid particle deposition on this surface, which we then placed in an oven at 70 °C for 3 h. In a previous study, a temperature of 70 °C and a reaction time of 3 h was favorable for improved crystallinity and photocatalysis. Prior to use, we thoroughly cleaned the glass substrates by ultra-sonication in acetone (99%) and performed hydrophilic treatment with a table-top UV/Ozone Processor (SSP16-110, SEN LIGHTS). We changed the concentration of Cu salts up to 10 mM for producing different doping levels. Here, we did not apply doping levels higher than 10 mM to avoid excess doping, which weakens photocatalytic activity. After the reaction proceeded for 3 h, we removed the substrate, which we rinsed in distilled water and dried with a blower. We obtained the powders after separation by centrifugation (3500 rpm) for 15 min, which we washed by dispersion in distilled water, centrifuged three times, and then dried at room temperature.

### 2.2. Characterization

We characterized the substrate samples with a Raman spectroscope (NRS-3300, JASCO), UV–vis spectrometry (UV mini 1240, Shimadzu), a scanning electron microscope (SEM) (JSM-7800F, JEOL), an electron probe micro-analyzer (EPMA) (JXA-8530F, JEOL), and X-ray photoelectron spectroscopy (XPS) (PHI5000 VersaProbe, ULVAC-PHI Inc.). We determined the optical band gap of the nanoparticles using diffuse reflection spectroscopy (DRS) with a fiber optic reflectance sphere (#58-583, Edmund OPTICS). We calculated the band gap energies using the Kubelka−Munk (K-M) method based on the diffuse reflectance spectra, where F(R) = (1 − R)^2^/2R [40].

### 2.3. Photocatalysis

We observed the photocatalytic activity of the Cu-TiO_2_ substrate through decomposing acetaldehyde gas. We performed the photocatalysis experiment based on a JIS R1701-2. We tested powder products for action spectral measurement through the degradation of methylene blue (MB, Nacalai Tesque, Inc., Nakagyo-ku, Kyoto, Japan) as a model organic compound. We prepared the solution for a reaction by adding 25 mg of the sample into 50 mL of 10 ppm MB solution in a glass beaker, which we then mixed via ultrasonication for several minutes. We initiated the photocatalytic reactions with xenon lamp (MAX505, Asahi Spectra) illumination, where we extracted the specific wavelength (320, 360, 400, 440, 480, and 520 nm) by using bandpass filters for action spectral measurement. During the reaction, we continuously stirred the solution with a magnetic stirrer at 300 rpm. After extracting the MB solution via centrifugation, we monitored the time-dependent decay of MB absorbance centered at 665 nm after 30, 60, 90, and 120 min of irradiation using a UV–Visible Spectrophotometer (Shimadzu UV mini-1240).

## 3. Results and Discussion

### 3.1. Choice of Cu Salts for Doping

Figure 2 presents the SEM images of undoped and doped samples prepared with different types of Cu salt: Cu(NO_3_)_2_·3H_2_O, CuCl_2_, CuSO_4_·5H_2_O, or Cu(CH_3_COO)_2_. In these experiments, we fixed the concentration of these salts at 10 mM. Most of the products had waxberry-like structures; in Figure 2, the size increased when using Cu salts. Among several types of Cu salts, the products with Cu acetate exhibited fewer protrusions, which was due to the inhibition of crystal growth. In addition, this morphology suggests poor photocatalytic activity due to is small surface area. Thus, crystal growth should be improved with the use of Cu nitrate or chloride, whereas Cu acetate is less effective in terms of forming fine structures.

To further understand the molecular vibrational state, we applied Raman spectroscopy. Figure 3 shows the Raman spectra of undoped and doped samples prepared with different types of Cu salt (10 mM). We assigned the Raman peaks at 152, 404, 513, and 634 cm^−1^ to the E_g_, B_1g_, A_1g_, and E_g_ modes of the anatase phase, respectively [39,41]. The strongest E_g_ vibrational mode (at 152 cm^−1^) seen in all samples was caused by symmetric O-Ti-O stretching vibrations in TiO_2_, indicating the formation of an anatase phase in the LPD process even with dissolving Cu salts in a precursor solution. The XRD patterns also confirmed the crystallographic nature of anatase TiO_2_, whereas the doped samples exhibited poor crystallinity comparing with the undoped ones (Appendix A). Based on the Scherrer equation, we calculated the crystallite size of undoped TiO_2_ to be ~50 nm; doped TiO_2_ is expected to be smaller according to the wide FWHM values [39,42]. Analyzing the XRD patterns of doped samples is difficult due to spectral noise; therefore, we performed curve-fittings in E_g_ Raman mode to evaluate the local structure of TiO_2_. Table 1 shows that the FWHM values increased with the addition of copper chloride and copper acetate; only the copper nitrate precursor solution had an FWHM value comparable to that of the undoped TiO_2_. These FWHM values could be explained by the structure of crystals. In general, the spectral line shape of crystal lattice vibrations (or phonons) is inversely proportional to the lifetime of the phonons [43]. In the case of an ideal harmonic crystal, the line shape is infinitesimally narrow. Experimental evidence shows that the Raman line always has a finite width, indicative of the presence of phonon-vibrational decay. The broadening of the Raman line width is attributed to the phonon scattering (decay) owing to the presence of impurities and/or defect sites and crystallite edges in the crystallites. Therefore, in our experiments, we expected the use of copper chloride and acetate to induce lattice disorders or size reduction to generate more impurities and/or defect sites and crystallite edges. However, copper nitrate, as shown in Table 1, produced fewer effects that hindered the atomic arrangement, resulting in maintaining the crystallinity of the products. Therefore, we concluded the use of Cu nitrate is more appropriate for Cu doping into TiO_2_ with LPD process from the viewpoint of improved crystallinity, which may enable efficient photocatalysis.

### 3.2. Concentration Dependence on Crystal Structures

We changed the concentration of Cu nitrate from 0.1 to 10 mM, and we observed the morphology of the powder products and coatings by SEM. Figure 4a–c and Figure 5a–d show SEM images of the powder and coating samples prepared with different concentrations of Cu nitrate (0.1, 1.0, and 10 mM). The size histogram measured with SEM is shown in Figure 6. The size of the particles was 1.4–1.5 µm, on average, for 0.1 and 1.0 mM. However, the particle size decreased as the concentration increased up to 10 mM. This was likely due to the excess doping, which inhibited crystallization.

For the coatings, we observed plate-like structures with large cracks between the plates on the glass substrate. The formation of large cracks between plates is an often-observed phenomenon when forming thick films [44]. The surface texture of a plate seemed to be smooth with the absence of protrusions, which considerably differed from that of the powder sample. We think this was caused by crystal growth from densely distributed nuclei. In the LPD process, nucleation starts from the hydroxyl groups, which act as nucleation sites. Without hydrophilization treatment of the substrate, we observed the decrease in density of nucleation sites, generating a waxberry-like structure with a sparse distribution (Appendix A).

We found that the change in Cu concentration affected the surface morphology of the Cu-TiO_2_ coatings, as shown in Figure 5b–d. The plate grew larger as the concentration increased from 0.1 to 1.0 mM. When we applied 10 mM, a granular structure formed, and plate-like structures did not form. We expected the addition of Cu ions into the precursor solution to enhance the crystallization to generate larger plates, which we observed for the produced powder samples (Figure 2). However, at the higher concentration (10 mM), plate-like structures and cracks were absent. The absence of cracks was due to the inhibition of nucleation and further crystal growth by the copper dopants dissolved in the solution. For the synthesis of TiO_2_ thin films, we employed an aqueous solution of ammonium hexafluoro titanate ((NH_4_)_2_TiF_6_) and boric acid (H_3_BO_3_) as the titanium fluoride complex and F^−^ scavenger, respectively. The titanium fluoride complex is hydrolyzed to titanium hydroxide and free F^−^ ions. The produced F^−^ ions can then be scavenged by H_3_BO_3_ [35,36,39]. The hydroxyl groups at the surface of a glass substrate induce hydrolyzation to form TiO_2_ films. When Cu nitrate is dissolved in a solution, Cu^2+^ is subject to diffusion over the hydroxylated surface [45]. Therefore, the diffusion of Cu ions over a hydroxylated surface is expected to diminish the hydrolysis (ligand-exchange) reaction of titanium fluoride complexes, resulting in a reduction in the density of TiO_2_ particles. The particles are isolated; therefore, less agglomeration occurs, so the cracks disappear.

Figure 7a,b represent the Raman spectra of powder and coating samples, respectively. Through the comparison of the spectral patterns in Figure 7a, we found that all samples were anatase-phase TiO_2_. Despite the weak peaks in the coating samples, as shown in Figure 7b, the presence of the principal E_g_ vibrational mode (at 155 cm^−1^) suggested the formation of anatase TiO_2_ on a substrate. Notably, the spectral width (FWHM) and peak wavenumber should be dependent on the Cu doping concentration.

Figure 8a,b depict the dependence of peak wavenumber and FWHM values on the concentration of Cu nitrate, respectively. In both the coating and powder cases, the strongest E_g_ peak blue-shifted as the Cu concentration increased from 0.1 to 10 mM. The blue shift in the E_g_ vibrational mode probably occurred due to the substitution of smaller Ti cations (0.605 Å) by larger Cu cations (0.73 Å). As larger Cu^2+^ ions replace smaller 4+ ones in their position, anatase TiO_2_ crystal is expected to be subject to compressive stress [46]. With higher doping concentrations, we observed a monotonical increase in the FWHM values of the E_g_ vibrational mode. The replacement of Cu by Ti ions was expected to induce disarrangement of Ti-O networks, which would result in the lower crystallinity of the products.

The EDS and EPMA elemental mapping images for undoped and Cu doped samples are presented in Figure 9 and Figure 10, respectively, which demonstrate that all the three elements (Ti, O, and Cu) were uniformly distributed in both the doped powder and coating samples. Thus, using LPD enables the preparation of doped TiO_2_ nanomaterials with a homogeneous distribution of doping ions.

### 3.3. Doping Levels and Optical Properties

We measured the atomic concentration of Cu in the powder and coating samples with EDS and EPMA; the results are shown in Figure 11. With increasing Cu nitrate concentration from 0 to 10 mM, the atomic concentration changed from 0 to 0.8 and 1.3 atm.% in the powder and coating samples, respectively.

Figure 12 presents the Kubelka–Munk (K–M) plots we obtained through diffusion reflectance measurements for the samples without (black) and with Cu doping (red, pink, and blue). We determined the bandgap of the material by extrapolating the linear part of [F(R)*hν]^1/2^ to zero when plotted against E(hν,) as shown in Figure 12. (see Table 2) The bandgap of undoped TiO_2_ powders is shown in Figure 12 at 3.30 eV, which was slightly larger than that of bulk TiO_2_ with an anatase phase (3.2 eV) [47]. As the crystallite size of the doped samples was less than 50 nm, the larger bandgap observed here might have been due to the particle size effect [48,49,50]. The subgap region showed an Urbach-like tail at 3.10–3.25 eV. This indicated a distribution of shallow states just below the conduction band minimum (CVM), though the origin of shallow gap states is still unclear [51]. When we applied 0.1 mM Cu nitrate, the bandgap remained at 3.30 eV; the tail observed with undoped TiO_2_ was invisible in the pink spectrum. The absence of the tail indicated less disorder (e.g., oxygen vacancy) formed in the TiO_2_ lattice, which agreed well with the results of SEM observation (Figure 5), Raman spectroscopy (Figure 8), and XRD analysis (Appendix A). In this case, with a single compensating oxygen vacancy, the empty 3d state of the Cu^2+^ ions lies above the CBM of the TiO_2_ host, so that the dopant does not impact the magnitude of the bandgap. With higher levels of Cu doping (1.0 and 10.0 mM), we observed two features, as shown in Figure 12: (i) red-shifting of the band originating from TiO_2_; (ii) creating additional states just below the original band. The red-shift of the bandgap to 3.15 eV (1.0 mM) and 2.90 eV (10.0 mM) could be explained by the coexistence of Cu^2+^ ions and oxygen vacancies. Mathew et al. studied the electronic states of Cu-TiO_2_ based on the density functional theory simulation, in which the introduction of Cu^+^ and Cu^2+^ ions by replacing Ti^4+^ ions in the TiO_2_ lattice created oxygen vacancies to extend the valence band maximum (VBM). Contrary to the case with 0.1 mM, Cu-derived states emerged between the valence band edge and extended the VBM to higher energies, leading to a decrease in the band gap. In addition, gap states were created between the VBM and CBM, which was evident as the presence of an absorption tail ranging from 3.05 to 3.25 eV for 1.0 mM and from 2.50 to 3.20 eV for 10.0 mM Cu nitrate. Cu-TiO_2_ with 10 mM Cu nitrate exhibited optical response in the visible region (wavelengths shorter than 430 nm), which is beneficial for the efficient use of solar light for photocatalysis.

### 3.4. XPS Analysis

Figure 13 presents the XPS spectra of coating samples synthesized with 10 mM Cu nitrate. Figure 13a,b indicate the narrow scans for Ti 2p and Cu 2p, respectively. We fitted the XPS data to highlight the peak position and FWHM values, which are summarized in Appendix A. The fitted curves are presented as dotted curves in Figure 13. We attributed the two strong symmetrical peaks at around 464.1 and 458.4 eV to Ti 2p_1/2_ and Ti 2p_3/2_, respectively. The peak separation of 5.7 eV shown in the Ti 2p doublet agrees well with the energy reported for TiO_2_ nanoparticles with an anatase phase [52]. Those peaks were 1.1 eV lower than those in anatase TiO_2_, which is generally caused by the presence of a higher anionic vacancy. The peak of the Ti 2p_3/2_ spectrum located between Ti^4+^ (459.2 eV) and Ti^3+^ (457.5 eV) [53]; however, we could not obtain a good fit with two components of Ti^4+^ and Ti^3+^, potentially due to the complex defect states of Cu/O vacancies. The Cu2p spectrum on deconvolution exhibited an intense peak at 932.4 eV and a shorter peak at 931.7 eV, which we ascribed to the Cu^2+^ and Cu^+^ states, respectively [31]. The absence of the satellite peak (942.6 eV) suggested that the Cu was doped in the TiO_2_ anatase lattice and did not form a surface CuO layer [54]. The doped Cu existing in oxidation state of Cu^+^ resulted in the formation of single oxygen vacancies. However, we did not observe the signal corresponding to Ti^3+^ sites in the Ti 2p region, potentially because that the Cu dopant present in the TiO_2_ lattice as either Cu^2+^/Cu^+^ had lower valency and higher electronegativity compared with those of Ti^4+^. To conserve the lattice charge as a result of the incorporation of aliovalent Cu ions in the anatase TiO_2_ matrix, ionic vacancies (O vacancies) in several sites may have formed.

### 3.5. Photocatalysis

We observed the photocatalytic activity of undoped and Cu doped TiO_2_ through decomposing acetaldehyde gas into CO_2_ and H_2_O. Figure 14 shows the concentration decay of acetaldehyde with light irradiation. In all cases, we confirmed the reduction of acetaldehyde gas and generation of CO_2_ (Appendix A) gas. Cu-TiO_2_ with 0.1 and 1.0 mM Cu nitrate exhibited a higher rate of reaction, whereas we found lower activity for the sample with 10 mM Cu nitrate doping. We calculated the average rate constants and standard deviations for undoped and Cu-doped TiO_2_, and the obtained values are summarized in Table 3. The Cu doping through the LPD process is described in Table 3, which increased photocatalytic efficiency (1.3 times). The undoped TiO_2_ prepared via LPD exhibited considerable MB degradation activity, as confirmed by a photocatalytic activity 3–4 times higher than that of the commercial photocatalyst P25-Degussa in our previous study [39]. Therefore, the photocatalytic activity of Cu-TiO_2_ in this study should be 4–5 times higher than that of the commercial photocatalyst P25-Degussa. Enhanced activity is most often driven by reduced charge carrier recombination in Cu-TiO_2_ materials [27,28]. This arises as the result of photogenerated electrons facilitating the reduction Cu^2+^ + e^−^ → Cu^+^, thus extending the valence band hole lifetimes at surfaces, which are able to react with adsorbed species to form active radicals. In our case, the results of XPS analysis confirmed the presence of Cu^2+^. The results of the Raman study and XRD analysis confirmed the absence of Cu_2_O structures. These results do not necessarily preclude the existence of amorphous or fine secondary oxide precipitates, which may improve exciton lifetime through electron capture in the secondary phase or enhance activity through an increased surface area [55,56,57]. The contribution of the mechanism discussed above was dominant in the 0.1 mM doped Cu-TiO_2_; an improvement in the light absorption of TiO_2_ resulted in the promotion of photocatalysis in the 1.0 and 10.0 mM cases. The light absorption edge extended to 3.15 and 2.90 eV for each sample, as shown in the K-M plots (Figure 12 and Table 2).

To reveal the contribution of the response in the visible light region, we evaluated the action spectra of Cu-TiO_2_ (10.0 mM). Figure 15 shows the light absorption and action spectrum of Cu-TiO_2_. Here, we applied 10 mM Cu nitrate, and we performed further annealing treatment to easily observe photocatalytic reaction. As shown in Figure 15, the action spectrum confirmed that the decomposition of MB occurred with 440 ± 10 nm or shorter wavelengths (corresponding to 2.76–2.88 eV energy). Thus, the Cu-TiO_2_ produced in our experiments responded to visible light to induce photocatalysis, which agrees with the bandgaps shown in Figure 12.

## 4. Conclusions

In this study, we found that after properly choosing a Cu dopant source for crystal growth, liquid phase deposition could be used produce Cu-doped TiO_2_ at low operating temperatures (around 70 °C) under atmospheric pressure, with a controllable level of incorporated ions. We dissolved several different types of Cu salts in a precursor solution as a source of Cu ions; we found that Cu nitrate was the best choice for generating Cu-doped TiO_2_ with improved crystallinity. The results of Raman spectroscopy revealed the formation of anatase TiO_2_ in all samples with and without Cu ion doping. Based on the SEM measurements, we observed that the structure of the powder samples was micron-sized particles with a waxberry-like structure, which maintained their structure after increasing the Cu nitrate concentration up to 10 mM. The coating samples produced with less than 1.0 mM displayed plate-like structure with cracks, whereas we found individually distributed particles by increasing the doping concentration to 10.0 mM. We think the plate-like structure formed due to the crystal growth from densely distributed nuclei. Additionally, excess doping of Cu ions inhibited the nucleation on the substrate to reduce the density of TiO_2_ particles, resulting in the disappearance of plates and cracks. The results of elemental mapping revealed the homogeneous distribution of Cu atoms across the doped particles and coatings, and we could control the doping level of Cu cations up to ~1.4 atm.% when we applied 10 mM Cu(NO_3_)_2_. Based on the results of the Raman and XPS studies, we concluded that the Cu ions were incorporated in the lattice of TiO_2_ in the Cu^+^ or Cu^2+^ states, which resulted in a slight shrinkage of the cell of the produced material. Finally, we enhanced the photocatalytic performance through decomposing acetaldehyde gas on Cu-TiO_2_ coating samples, which could be explained by their more efficient light absorption in the visible region or the efficient carrier diffusion in the products. Above a certain threshold (1.0 mM doping), increasing the Cu dopant concentration diminished the photocatalytic activity. We found that improvements in photocatalytic activity were optimized at Cu dopant levels between 0.2 and 0.4 atm.%. This approach is potentially applicable for coating visible light photocatalysts onto a technologically important substrate with a low melting-point, such as glass substrates, plastics, and polymers.

## Figures and Tables

**Figure 1 materials-16-00639-f001:**
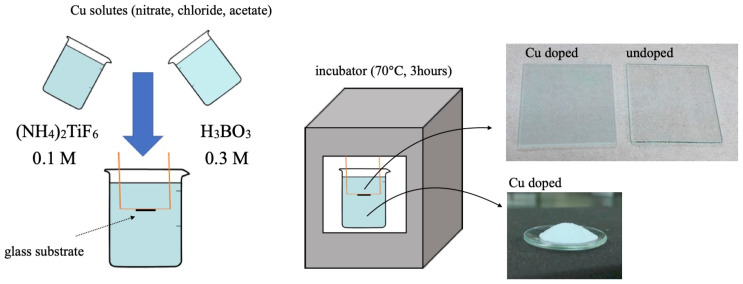
Schematic illustration of experimental procedure for producing Cu-doped TiO_2_ powders and coatings via liquid phase deposition.

**Figure 2 materials-16-00639-f002:**
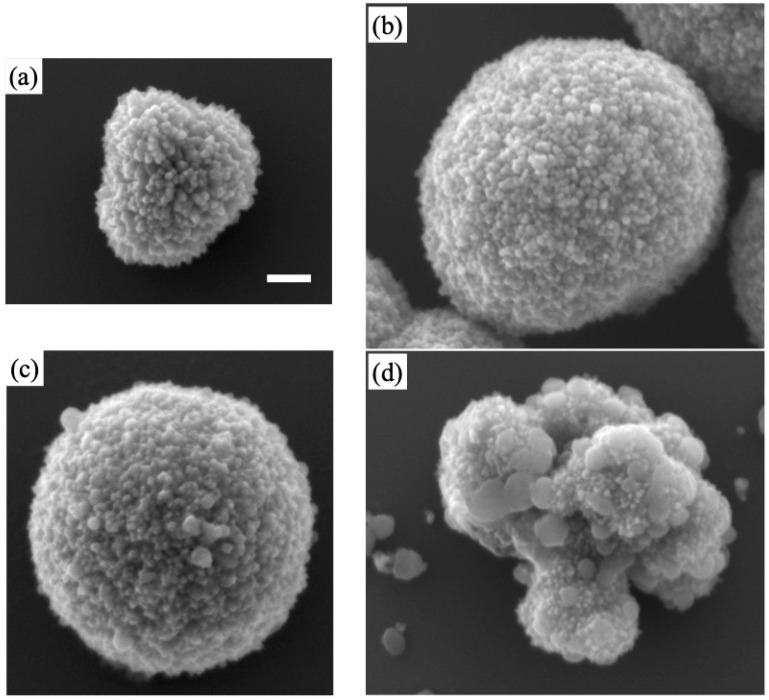
SEM images of (**a**) undoped and (**b**,**d**) Cu-doped TiO_2_ particles (10 mM): (**b**–**d**) nitrate, chloride, and acetate, respectively. Scale bar indicates 200 nm.

**Figure 3 materials-16-00639-f003:**
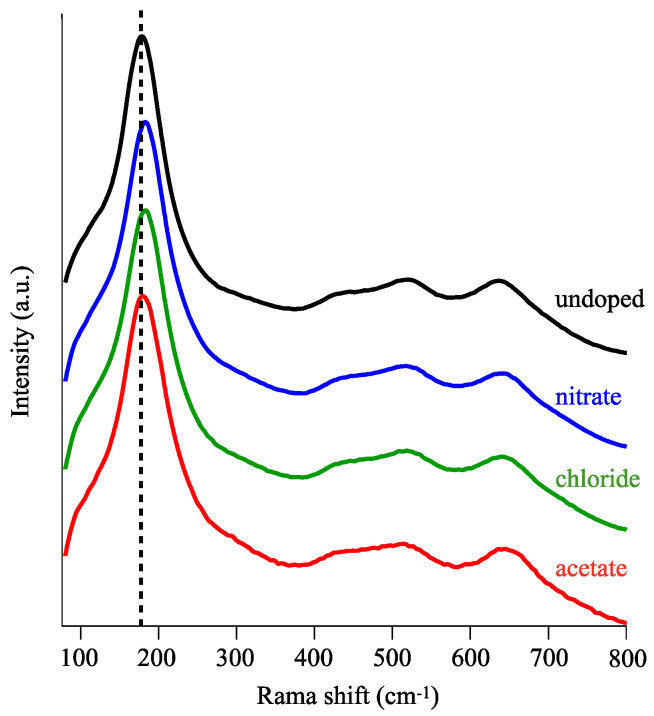
Raman spectra of undoped (black) TiO_2_ and TiO_2_ doped with nitrate (blue), chloride (green), and acetate (red). The spectral intensity was normalized.

**Figure 4 materials-16-00639-f004:**
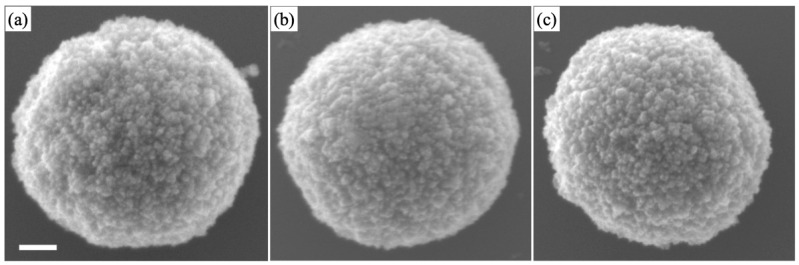
SEM images of Cu-doped TiO_2_ particles with different dopant concentrations: (**a**) 0.1, (**b**) 1, and (**c**) 10 mM. Scale bar indicates 200 nm.

**Figure 5 materials-16-00639-f005:**
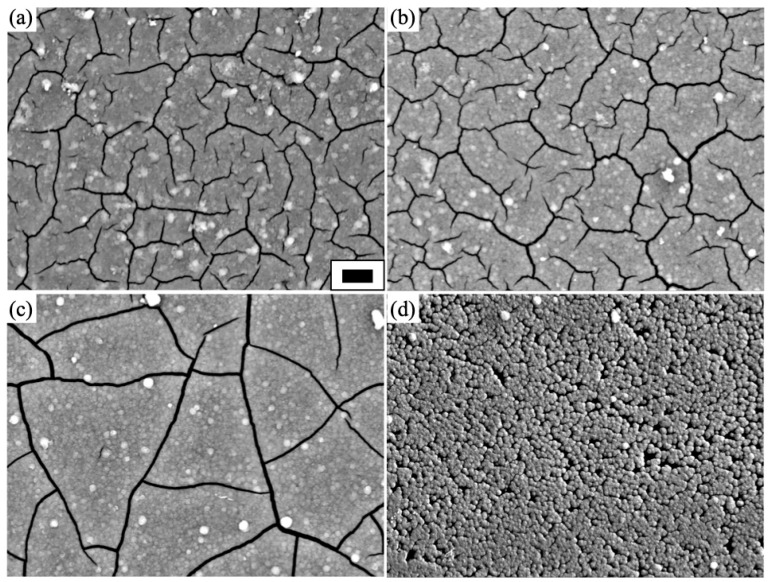
SEM images of Cu-doped TiO_2_ coatings with different dopant concentrations: (**a**) 0, (**b**) 0.1, (**c**) 1, and (**d**) 10 mM. Scale bar indicates 2 μm.

**Figure 6 materials-16-00639-f006:**
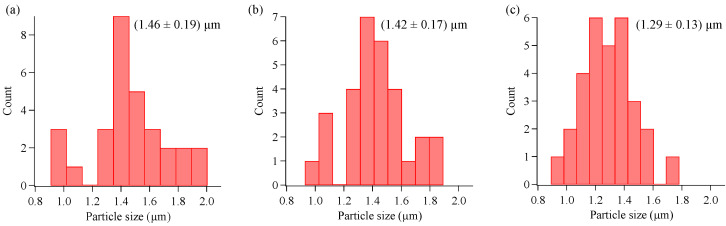
Size distribution of particles prepared with different concentrations of Cu nitrate: (**a**) 0.1, (**b**) 1, and (**c**) 10 mM.

**Figure 7 materials-16-00639-f007:**
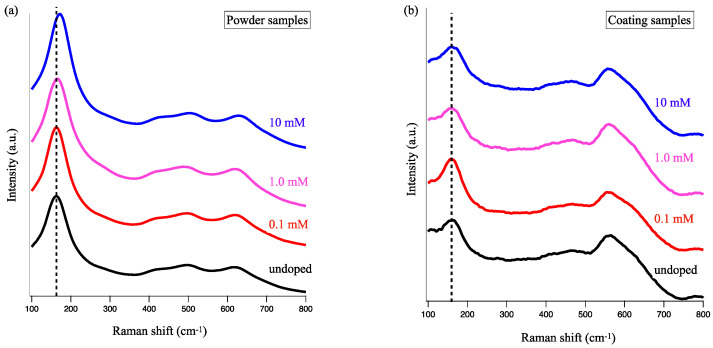
Raman spectra of Cu-TiO_2_ (**a**) powder and (**b**) coating samples. Black line indicates undoped TiO_2_. Red, pink, and blue lines correspond to Cu concentrations of 0.1, 1.0, and 10 mM, respectively.

**Figure 8 materials-16-00639-f008:**
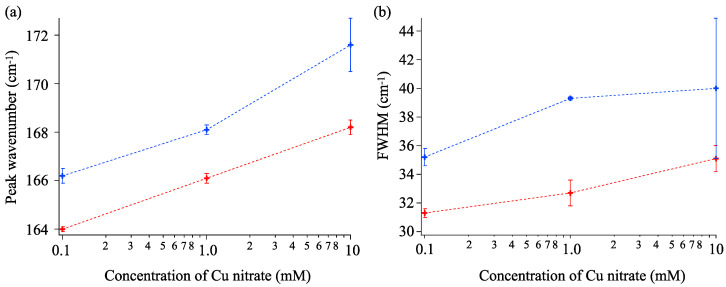
Concentration dependence on (**a**) peak wavenumber of Raman E_g_ mode and (**b**) spectral width. Red and blue indicate coating and powder samples, respectively.

**Figure 9 materials-16-00639-f009:**
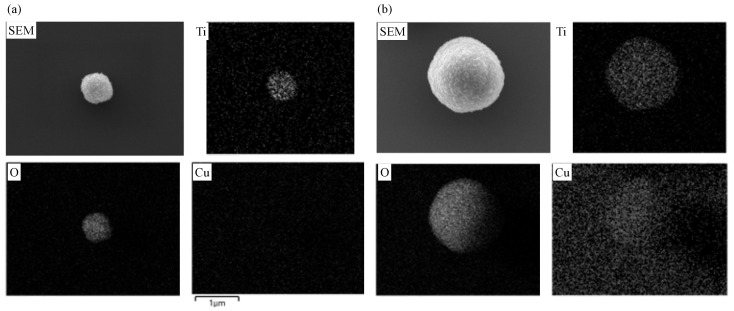
SEM and EDS mapping of (**a**) undoped and (**b**) Cu doped TiO_2_. Cu nitrate concentration was 10 mM.

**Figure 10 materials-16-00639-f010:**
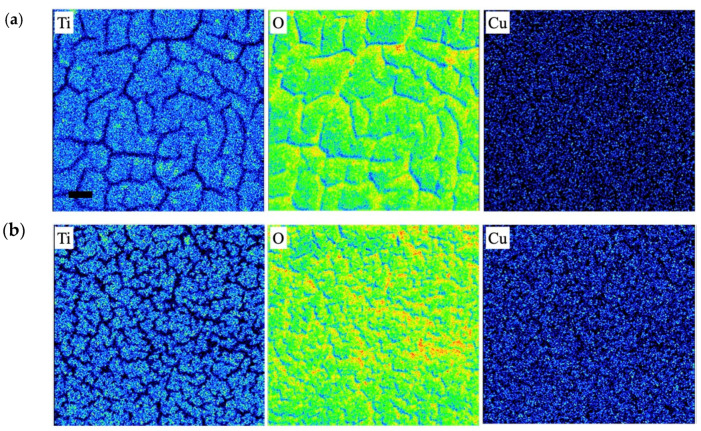
Elemental mapping of (**a**) undoped and (**b**) Cu doped TiO_2_ coated on glass substrate. Cu nitrate concentration was 10 mM.

**Figure 11 materials-16-00639-f011:**
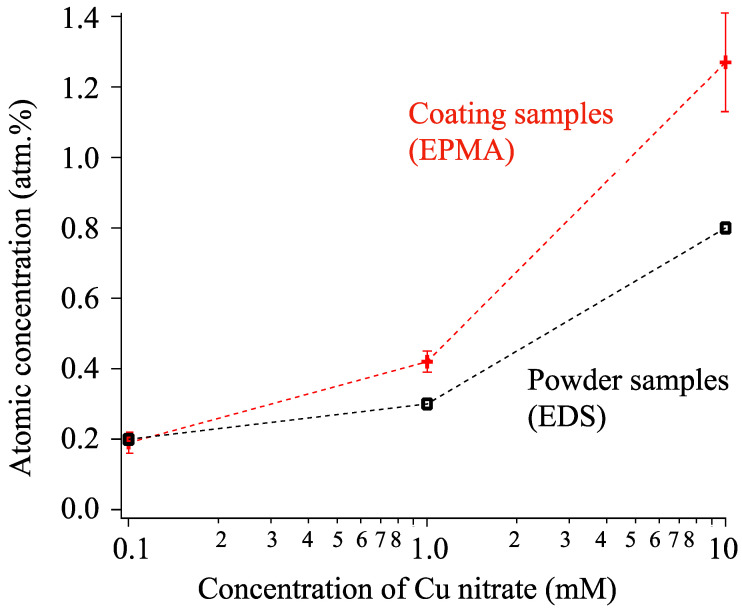
Atomic concentration of Cu incorporated into powder and coating samples.

**Figure 12 materials-16-00639-f012:**
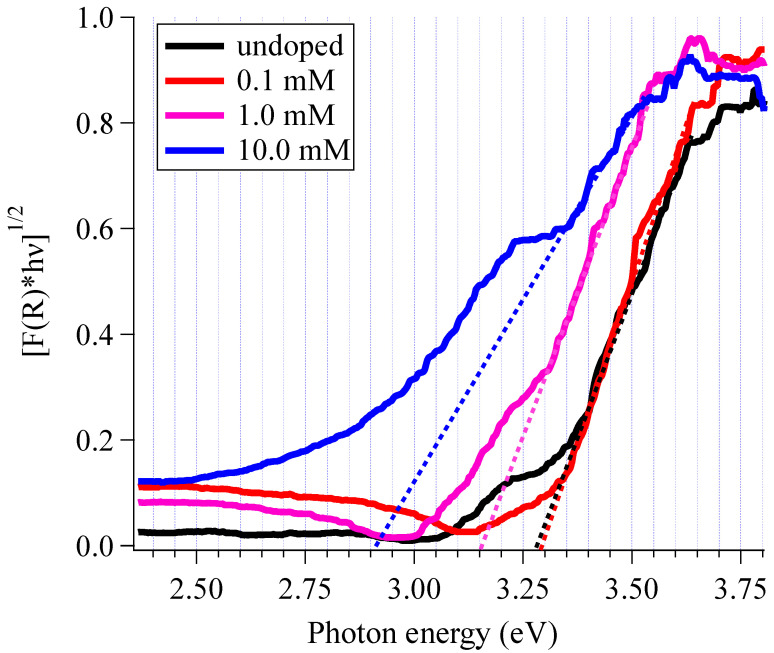
Tauc plots of samples prepared with 0 (black), 0.1 (red), 1.0 (pink), and 1.0 (blue) mM Cu nitrate.

**Figure 13 materials-16-00639-f013:**
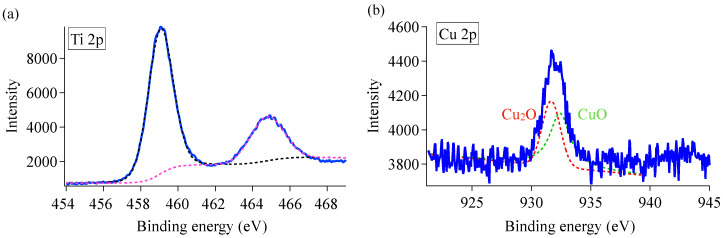
XPS spectra of Cu-doped TiO_2_ prepared with 10 mM Cu nitrate. (**a**,**b**) Narrow scans for Ti 2p and Cu 2p, respectively. (**b**) Green, red, and gray lines indicate Cu_2_O, CuO, and Cu(NO_3_)_2_, respectively.

**Figure 14 materials-16-00639-f014:**
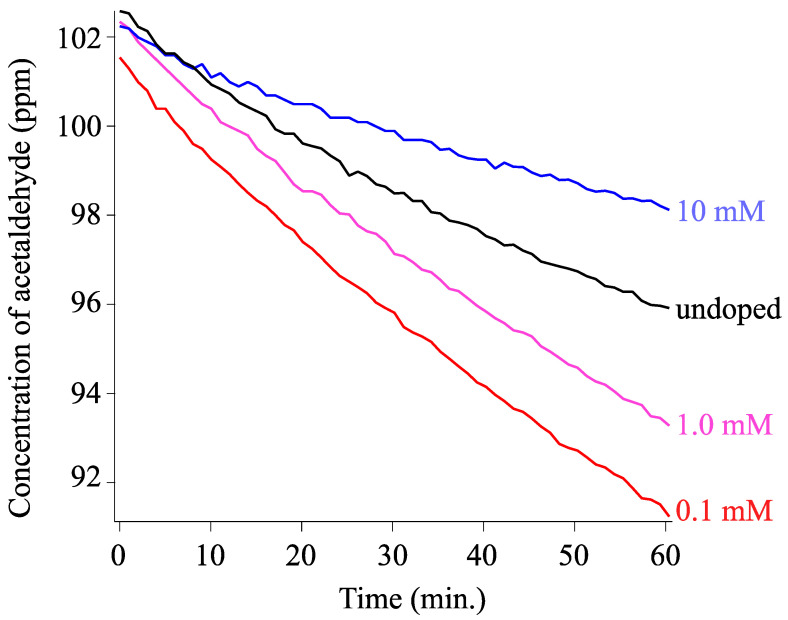
Photocatalytic decomposition of acetaldehyde by undoped (black) and Cu doped TiO_2_ (colors) coated on a glass substrate. Red, pink, and blue curves denote 0.1, 1.0, and 10.0 mM Cu nitrate used for synthesis, respectively.

**Figure 15 materials-16-00639-f015:**
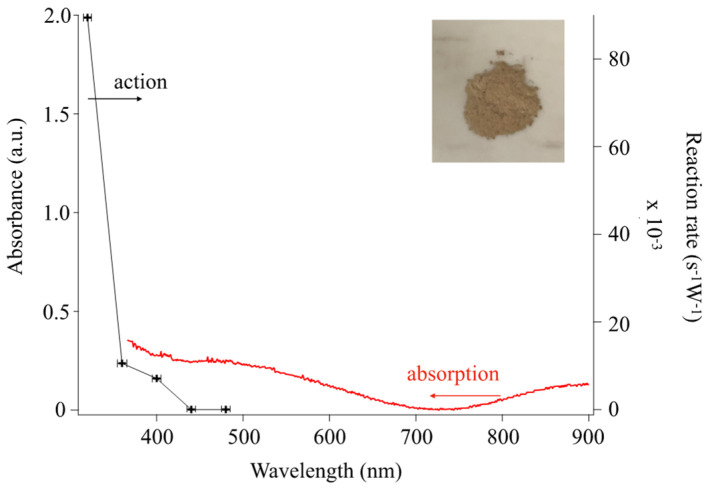
Action and absorption spectrum of Cu-TiO_2_ prepared with 10.0 mM of Cu nitrate. Thermal treatment under nitrogen for 1 h was performed after synthesis. Inset shows photograph of doped sample.

**Table 1 materials-16-00639-t001:** Dependence of FWHM values on Cu salt species.

Solute	FWHM (cm^−1^)
-	30.2
Cu(NO_3_)_2_	30.0
CuCl_2_	31.8
Cu(CH_3_COO)_2_	34.5

**Table 2 materials-16-00639-t002:** Concentration dependence on atomic concentration of Cu and bandgaps.

Conc. (mM)	Atm.%	Bandgap (eV)
0.0	0.0	3.30
0.1	0.2	3.30
1.0	0.4	3.15
10	1.3	2.90

**Table 3 materials-16-00639-t003:** Average rate constant for undoped TiO_2_ and Cu-TiO_2_.

Cu Nitrate (mM)	Rate Constant *k* (ppm/min.)	Std. Error (ppm/min.)
0.0	0.00131	0.00025
0.1	0.00164	0.00011
1.0	0.00156	0.00012
10	0.00068	0.00006

## Data Availability

Not applicable.

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
