# Peer review of "Low-Temperature Synthesis of Cu-Doped Anatase TiO2 Nanostructures via Liquid Phase Deposition Method for Enhanced Photocatalysis"

_materials, 2023, doi:10.3390/ma16020639_

Round 1
Reviewer 1 Report
- The author used the LPD method, may you make a small literature review with the other liquid-based methods of preparing the TiO2 and highlight the main advantages and disadvantages
- Can you provide also the XRD of Fig. 3 for a better understanding of the crystallinity issues
- The results of Table 1 should be explained more precisely
- In fig 6, why we do not see the same cracks in the (d) case
- In Fig 12, please start the y-axis from 0.5 and extend the x-axis more than 3.5ev, because I see that this curve is not correct due e the fact that the straight part is not so clear in the curves here. So please replot this curve
- The XPS data in Fig 13 b, should be fitted using any fitting software to highlight the position of the peaks
- The resolution of Fig 8 is not so good
Author Response
The authors are grateful to Reviewer for their constructive and thoughtful reading and comments. In the revised manuscript, we carefully followed all the comments and suggestions. In addition, English in the revised manuscript has been totally corrected through using English editing service of MDPI. Below, please find our point-to-point answers to all Reviewer’s comments.
Reviewer 1:
Comment 1: The author used the LPD method, may you make a small literature review with the other liquid-based methods of preparing the TiO2 and highlight the main advantages and disadvantages
Answer: As the referee suggested, we have added the statements related to nanoparticle preparation methods and the main advantages and disadvantages in line 45 and 50.
Comment 2: Can you provide also the XRD of Fig. 3 for a better understanding of the crystallinity issues
Comment 3: The results of Table 1 should be explained more precisely
Answer: The XRD patterns of undoped and Cu-doped TiO2 has been added as Fig. 1S in the supporting info. The diffraction patterns of all samples could be somehow assigned to an anatase phase of TiO2. The crystallinity of the Cu-doped TiO2 seems to be poor comparing to that of undoped one. Because signal to noise ratio is not large enough for data analysis, we have difficulty in comparing the crystallinity of doped ones. Therefore, Raman spectroscopy is needed to characeterize doped materials. Thanks to Reviewer's suggetion, more precise explanation of table 1 has been added in line 138.
Comment 4: In fig 6, why we do not see the same cracks in the (d) case
Answer: As you pointed out, the cracks are not observe in Fig.6(d). The reason for generating cracks is the agglomeration of densely distributed particles growing on a substrate. The absence of cracks might be due to the inhibition of nucleation and further crystal growth by copper dopants dissolved in the solution. For the synthesis of TiO2 thin films, an aqueous solution of ammonium hexafluoro titanate, (NH4)2TiF6, and boric acid, H3BO3, are employed as the titanium fluoride complex and F− scavenger, respectively. The titanium fluoride complex is hydrolyzed to titanium hydroxide and free F− ions. The produced F− ions can then be scavenged by H3BO3. The hydroxyl groups at the surface of a glass substrate induce hydrolyzation to form TiO2 films. When Cu nitrate is dissolved in the solution, Cu2+ is subject to diffusion over the hydroxylated surface. Therefore, the diffusion of Cu ions over the hydroxylated surface is expected to diminish the hydrolysis (ligand-exchange) reaction of titanium fluoride complexes, resulting in reducing the density of TiO2 particles. Since the particles are isolated, less agglomeration happens so that the cracks would disappear. Therefore, the sprace distribution of particles are found in Fig. 6(d). We have added the obove discussion in line 188.
Comment 5: In Fig 12, please start the y-axis from 0.5 and extend the x-axis more than 3.5ev, because I see that this curve is not correct due e the fact that the straight part is not so clear in the curves here. So please replot this curve
Answer: Since the light absorption by a glass substrate used in our experiments starts from 3.5 eV, the flactuation of spectral curves disturbs the analysis of bandgaps. To solve this problem, we carried out the diffusion reflectance measurement for powder samples, and the Taucs plots have been replaced to Kubelluca-Munk plots. Then, thanks to Reviewer's advice, the straight part becomes clear in the curves, and another finding has been provided. We have corrected the discussion of electronic states and bandgaps. (see line 239) Accordingly, table 2 was removed due to the behavior of complex electronic states.
Comment 6: The XPS data in Fig 13 b, should be fitted using any fitting software to highlight the position of the peaks
Answer: As Reviewer suggested, we performed fitting to obtain position, FWHM of the peaks. The result is summarized in Table S2, which has been added in supporting information. Disucussion about new findings via the analysis has been added in line 265.
Comment 7: The resolution of Fig 8 is not so good
Answer: Thank you for pointing it out. The resolution has been corrected.
Reviewer 2 Report
The manuscript addresses Low temperature synthesis of Cu-doped anatase TiO2 nanostructures via liquid phase deposition method for enhanced photocatalysis.
Introduction is too brief and it does mention direct applications.
The research gap is not clearly stated. There is no justification for the need of this research.
Methodology is not clearly explained. Section 2.1 is too brief. The concentration of Cu salts was changed up to 10 mM for making different doping levels, why?
Results are not compared with literature. Authors mention it is an enhanced photocatalytic process, where is the benefit?
Conclusions are too short and they do not mention any main findings either values from the results.
Formatting issues in many sections.
Author Response
The authors are grateful to Reviewer for their constructive and thoughtful reading and comments. In the revised manuscript, we carefully followed all the comments and suggestions. In addition, English in the revised manuscript has been totally corrected through using English editing service of MDPI. Below, please find our point-to-point answers to all Reviewer’s comments.
Reviewer 2:
Comment 1: Introduction is too brief and it does mention direct applications.
Comment 2: The research gap is not clearly stated. There is no justification for the need of this research.
Answer: As the referee pointed out, the statements related to the research gap is missing in the manuscript. We have added small literature review with the other liquid-based methods of preparing the TiO2 and highlight the main advantages and disadvantages for justification for the need of this research at line 45 and 50.
Comment 3: Methodology is not clearly explained. Section 2.1 is too brief. The concentration of Cu salts was changed up to 10 mM for making different doping levels, why?
Answer: Experimental studies have found improvements in photocatalytic activity to be optimized at Cu dopant levels between 0.1-2atm.%. Above a certain threshold, increasing Cu dopant concentration is often reported to bring about a diminishment in photocatalytic activity through a combination of enhanced recombination and shading. In our experiments, doping levels higher than 10 mM was not applied to avoid excess doping that diminish the photocatalytic activity. We have added the brief explanation of the reason why 10 mM was the highest in our experiments. Additionally, we have also added the explanation of experimental parameters (concentrations of precursor solution, temperature, reaction time).
Comment 4: Results are not compared with literature. Authors mention it is an enhanced photocatalytic process, where is the benefit?
Answer: We would like to compare with literature, but couldn't because of different experimental conditions such as light source (power, wavelength), substrate, reactor used for evaluating photocatalysis. The undoped products prepared via LPD exhibits considerable activity in terms of MB degradation, as it confirmed in a photocatalytic activity quite higher compared to that of the commercial photocatalyst P25-Degussa. The reaction rate of our undoped samples are 3 times higher than that of P25. Cu-doped TiO2 exhibited further improvement of activity (1.3 times). Therefore, by multiplying 3 by 1.3, the amplitude of increase in reaction rate corresponds to ~4 times comparing to the commercial photocatalyst P25-Degussa. We have added the statements that claims the photocatalytic activity higher than commercial photocatalyst P25, in line 288.
Comment 5: Conclusions are too short and they do not mention any main findings either values from the results.
Answer: Thank you for pointing it out. To mention our main findings, we have totally changed conclusion part. Please see new conclusions in the revised manuscript.
Comment 6: Formatting issues in many sections.
Answer: Thank you for pointing out the formatting issues. Formatting issues (font size, subscript etc.) have been corrected.
Reviewer 3 Report
A good piece of work made by authors however some clarification are needed as listed below:
The english should be slightly improved as many formulation were found weak such as “By Cu doping, the light absorption” also some other formulation were found the same
How sustainable is to produce this process at this cryogenics temperature “(<~80 ℃).” The authors contribution is not very well highlighted as they indicated different tools used for previous researchers used to make this
The scale bar in Figure 2 is misleading as it differ in sizes !
The font size for the graphs in Figure 8, 11,13, 14 should be improved as not visible now . They have to be consistent with Figure 15
Author Response
The authors are grateful to Reviewer for their constructive and thoughtful reading and comments. In the revised manuscript, we carefully followed all the comments and suggestions. In addition, English in the revised manuscript has been totally corrected through using English editing service of MDPI. Below, please find our point-to-point answers to all Reviewer’s comments.
Reviewer 3:
Comment 1: The english should be slightly improved as many formulation were found weak such as “By Cu doping, the light absorption” also some other formulation were found the same
Answer: English has been improved through using English editing sevice of MDPI.
Comment 2: How sustainable is to produce this process at this cryogenics temperature “(<~80 ℃).” The authors contribution is not very well highlighted as they indicated different tools used for previous researchers used to make this
Answer: To make clarify our contribution, we have added the statements related to nanoparticle preparation methods and the main advantages and disadvantages in line 45 and 50.
Comment 3: The scale bar in Figure 2 is misleading as it differ in sizes !
Answer: To avoid misleading, the scale bar has been adjusted to be same as 200 nm. Figure 2 was updated.
Comment 4: The font size for the graphs in Figure 8, 11,13, 14 should be improved as not visible now . They have to be consistent with Figure 15
Answer: Thank you for pointing out the invisibility of axis labels. We have corrected font size, and updated figures.
Round 2
Reviewer 2 Report
The paper has significantly improved and it should be published
Reviewer 3 Report
.